# Rethinking the Membrane Dynamics and Optimization Objectives of Spiking Neural Networks

**Hangchi Shen**[1,2]**, Qian Zheng**[3,4]**, Huamin Wang**[1,2,*]**, Gang Pan**[3,4]
[1]College of Artificial Intelligence, Southwest University
[2]Chongqing Key Laboratory of Brain Inspired Computing and Intelligent Chips
[3]The State Key Lab of Brain-Machine Intelligence, Zhejiang University
[4]College of Computer Science and Technology, Zhejiang University
`stephen1998@email.swu.edu.cn, hmwang@swu.edu.cn,`
`{qianzheng, gpan}@zju.edu.cn`

## Abstract

Despite spiking neural networks (SNNs) have demonstrated notable energy efficiency across various fields, the limited firing patterns of spiking neurons within fixed time steps restrict the expression of information, which impedes further improvement of SNN performance. In addition, current implementations of SNNs typically consider the firing rate or average membrane potential of the last layer as the output, lacking exploration of other possibilities. In this paper, we identify that the limited spike patterns of spiking neurons stem from the initial membrane potential (IMP), which is set to 0. By adjusting the IMP, the spiking neurons can generate additional firing patterns and pattern mappings. Furthermore, we find that in static tasks, the accuracy of SNNs at each time step increases as the membrane potential evolves from zero. This observation inspires us to propose a learnable IMP, which can accelerate the evolution of membrane potential and enables higher performance within a limited number of time steps. Additionally, we introduce the last time step (LTS) approach to accelerate convergence in static tasks, and we propose a label smooth temporal efficient training (TET) loss to mitigate the conflicts between optimization objective and regularization term in the vanilla TET. Our methods improve the accuracy by 4.05% on ImageNet compared to baseline and achieve state-of-the-art performance of 87.80% on CIFAR10-DVS and 87.86% on N-Caltech101. The code is available at `https://github.com/StephenTaylor1998/IMP-SNN`.

## 1 Introduction

In recent years, deep learning technology based on artificial neural networks (ANNs) has made significant breakthroughs in many fields [1, 2, 3, 4, 5]. However, as we all known, its high training and inference costs have become a major obstacle to restrict its further widespread applications. To overcome these challenges, a new generation of neural network architectures named spiking neural networks (SNNs) [6] have been developed, which may be a feasible path to refer to the efficient dynamics mechanism of biological nervous systems [7]. SNNs leverage the dynamic mechanisms of membrane potential integration and fire from biological neural networks, which can process time-varying input data by using a single model [8] as biological neurons, maintaining linearly increasing computational complexity. Therefore, the advantages of energy efficiency and biologically plausible dynamical mechanisms [7, 9, 10] make them a bridge between the fields of brain science and artificial intelligence, which is widely regarded as the next generation of ANNs [6]. In addition, it is worth mentioning that SNNs can cleverly avoid multiplication operations on neuromorphic chips

---

[*]Corresponding author.

38th Conference on Neural Information Processing Systems (NeurIPS 2024).

[11, 12, 13] by spike-based computing, achieving synaptic computation and membrane potential accumulation solely through addition operations, which can significantly enhance their computational efficiency.

It should be noted that training SNNs to achieve the comparable performance with the same ANN architecture remains a formidable challenge at present. The conversion method of ANNs to SNNs (ANN2SNN) [14, 15, 16, 17, 18] has proved to be an effective approach for obtaining high-performance SNNs. However, these methods mainly incorporate knowledge from the ANN's learning of static inputs into the converted SNNs, which motivates us to concern about the biological plausibility of these SNNs as they do not require the acquisition of spatiotemporal information. Another alternative is to leverage surrogate gradient [19, 20, 21] and backpropagation [22] through time to train high-performing SNNs. This method only requires four time steps to achieve an accuracy surpassing that of conversion methods requiring hundreds of time steps, making it the most promising training approach for currently available energy-efficient SNNs. Therefore, due to the development of direct training methods, SNNs have been extended to various tasks, including image classification[23, 24, 25], image reconstruction [26, 27, 28], object detection [29], natural language processing [30, 31], etc, and have demonstrated significant energy efficiency advantages in these fields. It is a pity that these training methods and architecture designs typically consider the firing rate or average membrane potential of the last layer as the output, instead of fully exploring the impacts of the membrane potential on the model performance and the training process [32, 33].

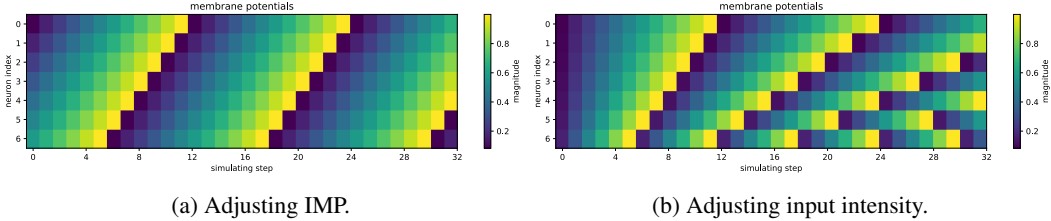

(a) Adjusting IMP.                                  (b) Adjusting input intensity.

Figure 1: Membrane potentials and spikes (yellow) generated by adjusting IMP and input intensity.

In this paper, we find that the initial membrane potential (IMP) has different effects on the membrane potential integration and spike firing compared to input intensity(Figure 1). Then, we discover that the novel spike patterns can be generated by adjusting the IMP of spiking neurons and additional mappings between firing patterns can be established, which motivates us to propose a learnable IMP. Furthermore, we find that in static tasks, the variation of the output accuracy at each time step for SNNs is directly related to the membrane potential, as it is the only time-varying term. Then we can explain why the convergence speed of temporal efficient training (TET) [34] in static tasks is significantly lower than standard direct training (SDT) [20]. To address this problem, we propose a method called last time steps (LTS) to achieve faster convergence in static tasks. And then, we propose a label-smoothed TET loss for neuromorphic tasks, which can outperform vanilla TET on neuromorphic datasets [35, 36]. It is worth noting that the proposed novel IMP method can obtain significant performance without any additional adjustments for the network structure and training method. By simply setting the IMP of the original LIF neurons to a learnable version, we achieve SOTA accuracies of up to 87.8% and 86.32% on CIFAR10DVS [35] and NCaltech101 [36], respectively. Furthermore, our LTS method improves the accuracy of SEW-ResNet50 [37] on the ImageNet1k [38] dataset to 71.83%, surpassing the vanilla SEW-ResNet152, 69.26%. The main contributions are as follows:

1) We discover that SNNs can generate new spike patterns by adjusting the IMP values, and prove that on the static tasks, the variation of SNN accuracy at each time step is only caused by the change of membrane potential. In addition, we innovatively introduce the learnable IMP in SNNs to accelerate the evolution of membrane potential.

2) To alleviate the slow convergence of TET on the static tasks, we propose the LTS method, which can accelerate the rate of convergence on the static tasks. Additionally, we construct a label-smoothed TET loss to further enhance the performances of SNNs on the neuromorphic tasks.

3) Compared with the baselines on the neuromorphic datasets and the large-scale static dataset ImageNet1k, our methods can achieve significant improvements. Moreover, there is almost no difference in the computational overhead and inference speed compared to the original models.

## 2  Related Works

### 2.1  Neuron Dynamics Modeling

The leaky integrate and analog fire spiking neuron [39] was proposed to replace binary spike with analog values for transmission, alleviating the issue of decreased performance in SNNs. The parametric leaky integrate-and-fire spiking neuron [40] was introduced to design a learnable dynamic model, enabling each neuron to learn optimal membrane time constants, thus increasing neuronal diversity. The gated leaky integrate-and-fire neuron [41] was employed for a channel-wise parameterization approach to fully parameterize the spiking neuron, including learnable decay mechanisms, potential thresholds, reset voltages, input conductance, and gating factors. The multi-level firing method [42] was used to enhance the expressive ability and achieve more efficient gradient propagation by integrating neurons with different thresholds to realize multi-level firing. Parallel spiking neurons [43] were discussed to remove the membrane potential reset process and redefine the dynamic mechanism in a non-iterative manner, addressing the difficulty of ordinary spiking neurons in learning long-term dependencies.

### 2.2  Direct Training Methods

The binary spikes emitted by spiking neurons during the forward phase are generated by a step function, which is a non-differentiable activation. In the backward propagation phase, this step function can be replaced with a surrogate gradient [19] to achieve direct training. The most common direct training method currently is backpropagation through time (BPTT) [22], which treats spiking neural networks (SNNs) as a special type of recurrent neural network (RNN). In this approach, gradients are propagated backward along the temporal dimension, which requires more computational resources and memory compared to their corresponding ANNs [44]. tdBN [21] explored normalization methods for spiking neural networks (SNNs) and achieved direct training of large-scale SNNs on the ImageNet dataset for the first time. Based on this, a more effective normalization method called TEBN [45] was proposed, which rescales the presynaptic inputs at each time step using distinct weights. TET [34] enabling spiking neural networks (SNNs) to converge to flatter minima compared to SDT [20], which enhances generalization capabilities. OTTT [46] and SLTT [47] simplified the gradient calculations along the temporal dimension in BPTT, significantly reducing memory and computational costs.

## 3  Analysis of Membrane Dynamics

In this section, we first investigate how the initial membrane potential affects neuronal spike patterns, and then analyze how the dynamic evolution of membrane potential drives improved SNN performance. These analyses underscore the critical role of membrane dynamics in SNNs and provide new insights into its impact on the SNN's representational capacity and convergence.

### 3.1  Preliminary of Spiking Neurons and Loss Functions

Spiking neurons are the fundamental unit of SNNs, used to simulate the dynamic behavior of brain neurons. Their operation is described by dynamical equations that are related to membrane potential and input current. The leaky integrate-and-fire (LIF) [7] model is one of the commonly used spiking neuron models, and its iterative form of the dynamical equations can be represented as follows:

$$h[t] = (1 - \tau)s[t] + I[t], \quad h \in \mathbb{R}^{T \times N}, \quad I \in \mathbb{R}^{T \times N} \tag{1}$$

$$o[t] = h[t] > V_{th}, \qquad o \in \{0,1\}^{T \times N}, \quad V_{th} \in \mathbb{R} \tag{2}$$

$$s[t+1] = h[t] - o[t], \qquad s \in \mathbb{R}^{T \times N}, \quad s[0] \in \{0\}^N \tag{3}$$

where $I[t]$ represents the neural input current at time $t$, $\tau$ represents the membrane potential decay coefficient. When $\tau = 0$, it reduces to the Integrate-and-Fire (IF) [7] neuron. $s[t]$ represents the state of the membrane potential at time step $t$, and $s[0]$ is the state of the IMP, which is typically set

to a constant 0. $h[t]$ represents the change in membrane potential during time step $t$, $o[t]$ indicates whether the neuron fires a spike at time $t$, and $V_{th}$ represents the firing threshold of the neuron.

The most commonly used loss functions in direct training SNNs are SDT [20] and TET [34]. The SDT loss function $\mathcal{L}_{\text{SDT}}$ is defined as:

$$\mathcal{L}_{\text{SDT}} = \mathcal{L}_{\text{CE}}(\frac{1}{T} \times \sum_{t}^{T} y[t], y_{gt}), \tag{4}$$

here, $T$ represents the total number of time steps, $y[t]$ represents the raw output of the model at each time step, $y_{\text{gt}}$ represents the ground truth label, and $\mathcal{L}_{\text{CE}}$ denotes the cross-entropy loss. SDT aggregate the outputs of the SNN by taking the mean of the outputs from all time steps, then calculate the loss based on the voting result. In TET, the averaging step is placed after the calculation of cross entropy loss:

$$\mathcal{L}_{\text{TET}} = \frac{1}{T} \times \sum_{t}^{T} \mathcal{L}_{\text{CE}}(y[t], y_{gt}), \tag{5}$$

the $\mathcal{L}_{\text{TET}}$ calculates the loss for each time step individually and then aggregates the losses from each time step to obtain the final loss. This approach can effectively improve the performance of SNN on neuromorphic datasets.

## 3.2 Membrane Dynamics Related to IMP

The membrane potential is commonly reset to zero before the next task in the current implementations of SNNs. However, through some experiments and the analysis of experimental results, we find that novel firing patterns and pattern mappings can be generated by adjusting IMP.

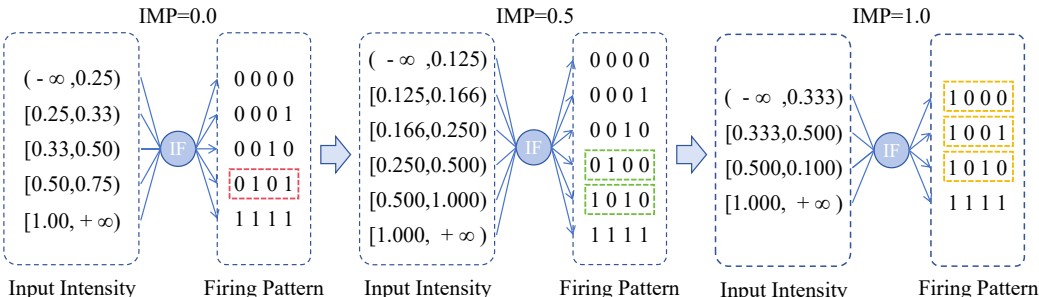

Figure 2: All firing patterns that IF neurons can generate under constant intensity input and 4 time steps. The red box highlights the disappearing firing patterns, while the green and yellow boxes denote the additional firing patterns due to the IMP change.

**Observation 1:** *Novel firing patterns under constant intensity input can be generated by adjusting IMP.* Figure 2 displays the firing patterns (spike sequences) generated by an IF neuron under constant input intensity. The number of patterns varies depending on the IMP. When the value of IMP increases from 0.0 to 0.5, the pattern '0101' disappears, while the new patterns '0100' and '1010' emerge, resulting in increased pattern diversity. Conversely, setting the IMP to 1.0 leads to the emergence of new patterns '1000', '1001', and '1010', but reduces the overall number of patterns. These additional patterns that cannot be generated if IMP is set to 0, can benefit static tasks by enabling SNNs to encode more information during the processing of static inputs.

**Observation 2:** *New mappings of firing patterns can be generated by adjusting IMP.* Apart from its impact on input encoding, our primary focus is whether the model's capabilities can be enhanced by modifying IMP. In ANNs, artificial neurons can map any single input variable to any value by adjusting the weights. Similarly, we hope that spiking neurons can also map input sequences to as many firing patterns as possible, thereby enabling the network to have better representation capabilities. From figure 3, we observe that every output pattern has at least one available pattern mapping. However, we can notice that black areas present in the figure, which means that no matter how the synaptic weights are adjusted, mapping among these spike patterns still cannot be established. In other words, spiking neurons can theoretically generate all firing patterns, but they cannot map any

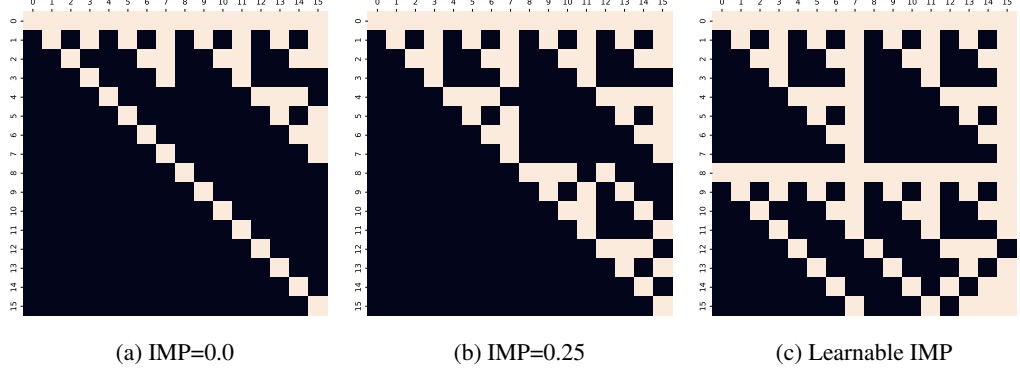

| (a) IMP=0.0 | (b) IMP=0.25 | (c) Learnable IMP |

Figure 3: Pattern mapping of IF neuron over 4 time steps. The horizontal and vertical axes in the figure represent all possible spike patterns (16 total) that IF neurons may receive and emit. The white squares indicate that IF neuron can receive the spike pattern from the horizontal axis and emit a spike pattern on the vertical axis, known as pattern mapping.

specific pattern to all patterns. Figure 3b illustrates that through adjusting the IMP value from 0 to 0.25 can generate additional pattern mappings. Furthermore, Figure 3c demonstrates that when IMP is learnable, it exhibits a greater potential for establishing mappings among spike patterns. Therefore, we believe that the learnable IMP can effectively improve the expression capacity of spiking neurons.

### 3.3 Membrane Potential Evolution in Static Tasks

We further explore the effect of membrane potential variations on static tasks through a image classification task. We define SNN as follows to focus on its performance at each time step,

$$(s[t+1], y[t]) \leftarrow f(x[t], s[t], \theta), \tag{6}$$

where $f(\cdot)$ represents the network computation, $\theta$ represents the network weights, $s[t]$ represents the set of membrane potentials of all neurons in SNN with time step $t$, $x[t]$ represents the input at time step $t$, and $y[t]$ is the corresponding output. Assuming $x = x(t)$ is a constant input intensity for $t = 1, 2, ..., T$, then we can simplify Eq. 6 as:

$$y[t] = f(x, s[t], \theta). \tag{7}$$

On the static tasks, the temporal variations are determined solely by the state of the membrane potential $s[t]$, resulting in corresponding changes for the output $y[t]$.

**Observation 3:** *In static tasks, the accuracy of SNNs at the each time step is sensitive to the current MP.* Figure 4 demonstrates the test accuracy of SNNs at each time step on cifar10 dataset [48], which shows that the accuracy is extremely low at T=1, only $10.76\%$. However, as the time step T increases, the model accuracy exhibits an upward trend, exceeding $90\%$. It is worth noting that on the static tasks, since the input $x$ and weight $\theta$ are fixed at each time step, the model's output at each time step is entirely determined by the current state of the membrane potential $s[t]$. For instance, when the membrane potential evolves to a "sufficiently good" state, such as at $t = 4$, the SNNs only requires the current membrane potential $s[4]$ and input $x$ to achieve an accuracy of $92.05\%$, which is close to the model's final performance of $92.36\%$. Therefore, these findings prompt us to reconsider how to accelerate the evolution of the membrane potential to enhance the SNNs performances within a limited number of time steps.

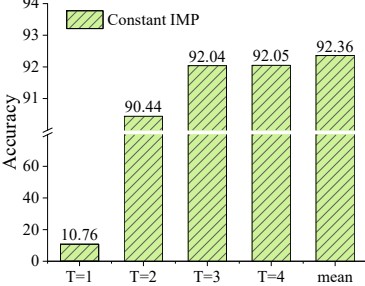

Figure 4: The test accuracy at each time step on the CIFAR10 dataset.

**Observation 4:** *TET performs well on the neuromorphic tasks but exhibits slow convergence on the static tasks.* We compare the SDT loss and the TET loss on the static datasets and the neuromorphic

Table 1: Test accuracy of TET and SDT on the static and neuromorphic datasets.

| Loss Function | Static Dataset(SEW-R18) | | | Neuromorphic Dataset(VGG11) | | |
|---|---|---|---|---|---|---|
| | CIFAR10/100 | ImageNet100 | ImageNet1k | CIFAR10DVS | DVSG128 | NCaltech101 |
| SDT Loss | **94.56/76.58** | **78.42** | **63.21** | 84.3 | 98.26 | 85.78 |
| TET Loss | 94.33/76.40 | 77.80 | 62.92 | **85.6** | **98.61** | **86.32** |

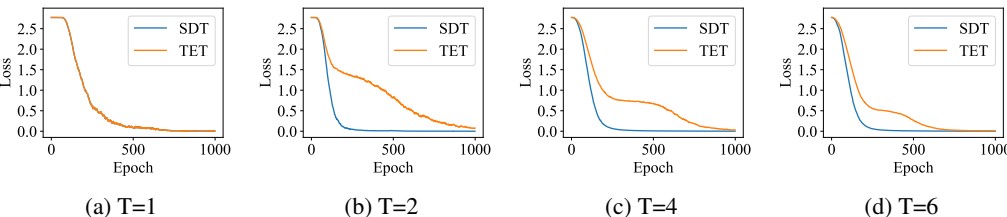

(a) T=1       (b) T=2       (c) T=4       (d) T=6

Figure 5: The convergence speed of TET and SDT on the static data.

datasets, and find that TET loss has a significant advantage on the neuromorphic datasets, but it is not superior to SDT loss on the static datasets, as shown in Table 1. We believe that this phenomenon arises from the constant intensity input on the static datasets. By applying the TET loss Eq. 5 to the SNNs with the static input Eq. 7, we have:

$$\mathcal{L}_{\text{TET}} = \frac{1}{T} \times \sum_{t}^{T} \mathcal{L}_{\text{CE}}(y[t], y_{gt}), \; where \; y[t] = f(s[t], x, \theta). \tag{8}$$

It can be observed that on static tasks, the membrane potential $s[t]$, as the only time-varying term in the dynamic system, evolves gradually as the time step progresses. However, according to **Observation 3**, we know SNNs are sensitive to $s[t]$, which means that it is difficult to output the same result for different $s[t]$. The optimization goal of the TET loss is to make the network output the correct results for every time step, i.e. $y[t] = y_{gt}$. In this case, TET requires more iterations to build a flat "landscape", slowing down the convergence, as shown in Figure 5. Additionally, if the SNNs are insensitive to the value of $s[t]$ and even tends to the same value for all time steps, i.e. $y[1] = y[2] = ... = y[T] = \frac{1}{T} \sum_{t}^{T} y[t]$, then the output of the first time step is sufficient to represent the results of all T time steps. In this case, the computations for the subsequent time steps become redundant and meaningless.

## 4 Methods

### 4.1 Learnable IMP

The membrane potential in the current dynamic model of SNNs is typically initialized to a uniform constant value (usually 0). Based on Observation 1 and Observation 2, we find that the learnable IMP can enhance the expressive power of SNNs. Based on Observation 3, the learnable IMP can allow the membrane potential to start from a non-zero value, which may help improve the performances of SNNs on the static tasks. We can assign an independent learnable IMP for each spiking neuron. According to Eq. 1, the membrane potential accumulation can be represented as follows,

$$h[t] = \hat{s}[t] + x[t], \quad x \in \mathbb{R}^{T \times N}, \; h \in \mathbb{R}^{T \times N}, \; \hat{s} \in \mathbb{R}^{T \times N}, \; \hat{s}[0] \in \mathbb{R}^{N}, \tag{9}$$

where $\hat{s}[0]$ represents the state of the IMP that is extended from zero to the real number. As a counterpart to the method of setting IMP to 0, in the initialization process, we set IMP to a uniform distribution with an expected value of 0,

$$\hat{s}[0] = Uniform(-\lambda, \lambda)^{N}, \tag{10}$$

where $\lambda$ is a hyper-parameter used to control the boundaries of the uniform distribution, ensuring the IMP remains within an appropriate range. Since the current implementation of SNNs requires memory allocation to store membrane potential states, replacing 0 with a trained IMP during the inference process will not incur additional computational overhead.

## 4.2 LTS Method for the Static Tasks

We propose a new post-processing representation method called last time step (LTS) to alleviate the convergence difficulties of TET on the static tasks, based on Observation 3 and Observation 4. This approach masks all outputs of the SNNs before the LTS and only retains the output of the LTS as the result of the entire model, which ensures that the SNNs can generate the most "high-quality" membrane potential without interference before the LTS $T$,

$$y[T] = f(s[T], x, \theta). \tag{11}$$

where $y[T]$ represents the LTS's output of the SNNs. When using only the LTS as the output, both SDT and TET losses yield the same representation,

$$\mathcal{L}_{\text{LTS}} = \mathcal{L}_{\text{CE}}(y[T], y_{gt}). \tag{12}$$

## 4.3 Label Smooth TET Loss for the Neuromorphic Tasks

Based on the results in Table 1, we recommend using TET loss to achieve better performances when dealing with the neuromorphic tasks. In the original version of TET [34], an additional mean squared error (MSE) regularization term was added to control the firing level of the last layer of the model, given by $\mathcal{L}_{\text{REG}} = \frac{1}{T} \sum_{t=1}^{T} \mathcal{L}_{\text{MSE}}(y[t], \phi)$, where $\phi$ denotes the target firing level. The coefficient $\lambda$ controls the proportion of the two losses, and the complete loss is defined as $\mathcal{L}_{\text{Total}} = (1 - \lambda)\mathcal{L}_{\text{TET}} + \lambda\mathcal{L}_{\text{REG}}$. We think that this setup will prevent the training loss of the model from converging to zero, because when $\mathcal{L}_{\text{REG}} = 0$, the model will output a constant value 1 at every time steps, rendering the model unable to perform the classification task. On the other hand, when $\mathcal{L}_{\text{TET}} = 0$, $\mathcal{L}_{\text{MSE}} > 0$. Considering that $\mathcal{L}_{\text{MSE}}$ plays a role similar to a smoothing process in the loss function, we propose removing $\mathcal{L}_{\text{REG}}$ and replacing the cross-entropy loss with a label smooth cross-entropy loss, as shown by the following equation:

$$\mathcal{L}_{\text{TET-S}} = \frac{1}{T} \times \sum_{t}^{T} \mathcal{L}_{\text{CE}}(f(s[t], x, \theta), \hat{y}_{gt}), \; where \; \hat{y}_{gt} = (1 - \epsilon)y_{gt} + \frac{\epsilon}{K}, \tag{13}$$

here, $y_{gt}$ represents the ground truth, $\hat{y}_{gt}$ represents the smoothed label, $\epsilon$ represents the smoothing factor, and $K$ represents the number of classes. It can be observed that $\mathcal{L}_{\text{TET-S}}$ can effectively avoid the trade-off between firing level and classification accuracy for model training, and can theoretically allow the training loss to converge to zero.

## 5 Experiments

In this section, we demonstrate the effectiveness of our proposed method by extensive experiments. We compare the results of our method with other methods on both the neuromorphic dataset and the static dataset. Additional training procedures and other hyperparameter settings are provided in the appendix A.

### 5.1 Execution Speed Benchmark of IMP

We compare the execution speed and the memory consumption between the vanilla IF neurons and IF+IMP neurons in Figures 6. The number of neurons are set to $2^8$, $2^{12}$, $2^{16}$, and $2^{20}$, with time steps of 2, 4, 8, 16, and 32. All neurons are implemented by using spikingjelly and Py-Torch, and the computations are performed on GPU. It can be observed that there is almost no difference (about $\pm 1.03\%$) in the execution time between the IF neurons with IMP and the vanilla IF neurons, including forward and backward propagation. In addition, since the computational consumption of SNNs is mainly caused by synaptic computation, the additional overhead caused by adding IMP can be neglected.

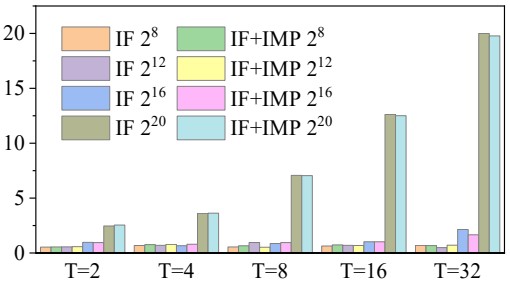

Figure 6: Execution time (ms) for the forward and backward pass of IF neurons, w/wo IMP.

## 5.2 Convergence Speed of LTS on the Static Data

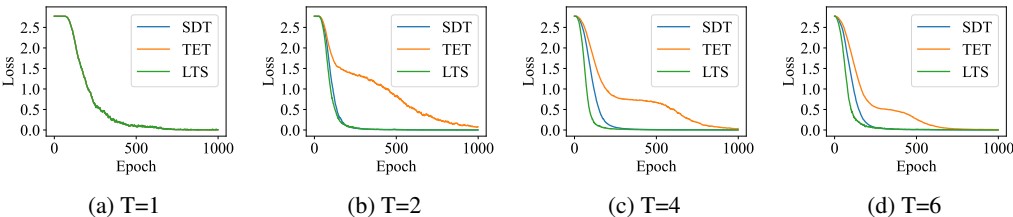

|          |          |          |          |
|:--------:|:--------:|:--------:|:--------:|
| (a) T=1  | (b) T=2  | (c) T=4  | (d) T=6  |

Figure 7: Convergence speed of LTS, TET, and SDT on the static data.

We have conducted a validation of the convergence speed of SDT, TET, and LTS on the commonly used time steps (1, 2, 4, and 6), as shown in Figure 7. The application of LTS post-processing has resulted in an improvement in the convergence speed of SNNs.

## 5.3 Performances on the Neuromorphic Data Classification

Table 2: Comparison of our methods and other SOTA methods on the neuromorphic datasets. Size refers to the input resolution of SNNs.

| Dataset | Method | SNN Architecture | Size | Time Steps | Accuracy(%) |
|---|---|---|---|---|---|
| | GLIF[41] | Wide 7B Net | 48 | 16 | 78.10 |
| | NDA[49] | VGG | 48 | 10 | 79.60 |
| | TET[34] | VGG | 48 | 10 | 83.17 |
| | TEBN[45] | VGG | 48 | 10 | 84.90 |
| | PSN[43] | VGG | 48 | 10 | 85.90 |
| | IMP(ours) | VGG | 48 | 10 | **85.90** |
| | IMP+TET-S(ours) | VGG | 48 | 10 | **87.10** |
| | IMP+TET-S(ours) | VGG | 48 | 8 | **87.80** |
| CIFAR10-DVS | PLIF[40] | PLIF Net | 128 | 20 | 74.80 |
| | TDBN[21] | ResNet-19 | 128 | 10 | 67.80 |
| | Dspike[50] | ResNet-18 | 128 | 10 | 75.40 |
| | KLIF[51] | PLIF Net | 128 | 15 | 70.90 |
| | SEW ResNet[52] | Wide 7B Net | 128 | 16 | 74.40 |
| | Spikformer[23] | Spikformer | 128 | 10 | 78.90 |
| | Spikformer[23] | Spikformer | 128 | 16 | 80.90 |
| | NDA[49] | VGG | 128 | 10 | 81.70 |
| | IMP(ours) | VGG | 128 | 16 | **86.30** |
| | IMP+TET-S(ours) | VGG | 128 | 16 | **87.00** |
| | NDA[49] | VGG | 48 | 10 | 78.20 |
| | EventMix[53] | ResNet18 | 48 | 10 | 79.47 |
| | ESP[54] | SNN7-LIFB | 48 | 10 | 81.74 |
| | TCJA[55] | TCJA-SNN | 48 | 10 | 82.50 |
| | TKS[56] | VGG-TKS | 48 | 10 | 84.10 |
| | IMP(ours) | VGG | 48 | 10 | **84.68** |
| N-Caltech101 | IMP+TET-S(ours) | VGG | 48 | 10 | **85.01** |
| | EventDrop[57] | VGG | 128 | 10 | 74.04 |
| | NDA[49] | VGG | 128 | 16 | 83.70 |
| | EventRPG[58] | VGG | 128 | 10 | 85.62 |
| | STR[59] | VGG | 128 | 10 | 85.91 |
| | IMP(ours) | VGG | 128 | 16 | **86.12** |
| | IMP+TET-S(ours) | VGG | 128 | 16 | **87.86** |

We have applied our methods to a simple spiking VGG model and compare them with the SOTA SNNs on the neuromorphic datasets. Since the CIFAR10DVS and NCaltech101 datasets are not pre-divided into training and testing sets, we split these datasets in a 9:1 ratio. To ensure a fair

comparison with other existed methods, we adopt the two configurations with resolutions of 48 and 128, respectively. The data preprocessing and training settings refer to the appendix A.8.

Table 2 reports the experimental results on the CIFAR10DVS and NCaltech101 datasets. On the CIFAR10DVS dataset, when setting the resolution to 48, the IMP method achieves the SOTA accuracy of 85.9%, which is 2.73% higher than the baseline [34] and is on par with the current SOTA method PSN. It should be noted that the removal of the reset process in PSN means that the spiking activities in the previous time steps will not affect the membrane potential values in the subsequent time steps. When we set the resolution to 128, the IMP method once again demonstrates its superiority, achieving the SOTA accuracy of 86.3%, which exceeds all other methods, including the current SOTA data augmentation method EventRPG[58]. In addition, we further explore the impact of using smoothed TET loss on model performance. The experimental results show that the performance of the model has been significantly improved under both configurations, achieving accuracies of 87.00% and 87.10% respectively. In addition, the accuracy can be further improved to 87.8% by setting the time step to 8. On the NCaltech101 dataset, our method can also demonstrate excellent performance. Specifically, when we set the resolution to 48, our method achieves the current SOTA accuracy of 85.01%. When switching to resolution of 128, our method further demonstrates its advantages, achieving the SOTA accuracy of 87.86%.

## 5.4 Performances on the Static Data Classification

Table 3: Comparison of our methods and other methods on the ImageNet1k dataset.

| Method | Network Architecture | Reset | Params | Time Steps | Accuracy(%) |
|---|---|---|---|---|---|
| PSN[43] | SEW ResNet-18 | ✗ | 11.69 | 4 | 67.63 |
| | SEW ResNet-34 | ✗ | 21.79 | 4 | 70.54 |
| Dspike[50] | ResNet-34 | ✓ | 21.79 | 6 | 68.19 |
| | VGG-16 | ✓ | 138.42 | 5 | 71.24 |
| TET[34] | SEW ResNet-34 | ✓ | 21.79 | 4 | 68.00 |
| TDBN[21] | ResNet-34 | ✓ | 21.79 | 6 | 67.05 |
| TEBN[45] | SEW ResNet-34 | ✓ | 21.79 | 4 | 68.28 |
| GLIF[41] | ResNet-34 | ✓ | 21.79 | 4 | 67.52 |
| Spikformer[23] | Spikformer-6-512 | ✓ | 23.37 | 4 | 72.64 |
| | Spikformer-8-512 | ✓ | 29.68 | 4 | 73.38 |
| SEW ResNet[52] | SEW ResNet-18 | ✓ | 11.69 | 4 | 63.18 |
| | SEW ResNet-34 | ✓ | 21.79 | 4 | 67.04 |
| | SEW ResNet-50 | ✓ | 25.56 | 4 | 67.78 |
| | SEW ResNet-101 | ✓ | 44.55 | 4 | 68.76 |
| | SEW ResNet-152 | ✓ | 60.19 | 4 | 69.26 |
| LTS | SEW ResNet-18 | ✓ | 11.69 | 4 | 64.33(+1.15) |
| | SEW ResNet-34 | ✓ | 21.79 | 4 | 68.10(+1.06) |
| | SEW ResNet-50 | ✓ | 25.56 | 4 | 71.24(+3.46) |
| IMP+LTS | SEW ResNet-18 | ✓ | 14.17 | 4 | **65.38**(+2.20) |
| | SEW ResNet-34 | ✓ | 25.54 | 4 | **68.90**(+1.86) |
| | SEW ResNet-50 | ✓ | 36.67 | 4 | **71.83**(+4.05) |

We apply our proposed IMP and LTS post-processing methods to the standard SEW-ResNet[52] architecture and compare them with the SOTA spiking neurons and the directly training SNN methods on the large-scale static dataset ImageNet1k [38].

Table 3 presents the detailed experimental results on the large-scale static dataset ImageNet1k. Specifically, by applying the LTS post-processing method to the SEW-ResNet18/34/50 models, we can obtain the accuracy improvements of 1.15%, 1.06%, and 3.46% compared to the baselines, respectively. These results demonstrate the effectiveness of LTS on the large-scale datasets. Furthermore, with the introduction of the learnable IMP, the accuracy can be further increased by 2.2%/1.86%/4.05%. With

the LTS post-processing and learnable IMP, our SEW-ResNet50 achieves an accuracy of 71.83%, surpassing the accuracy of the vanilla SEW-ResNet152, 69.26%.

## 5.5 Further Ablation Studies

Table 4: Ablation Study on CIFAR10DVS and Imagenet100.

| Dataset | Method | Spiking Network | Time-steps | Accuracy(%) |
|---------|--------|-----------------|------------|-------------|
| CIFAR10-DVS | SDT($\epsilon = 0.0$) | VGG | 10 | 83.70 |
| | TET($\epsilon = 0.0$) | VGG | 10 | 84.90 |
| | TET-S($\epsilon = 0.1$) | VGG | 10 | 85.60 |
| | TET-S($\epsilon = 0.01$) | VGG | 10 | **86.10** |
| | TET-S($\epsilon = 0.001$) | VGG | 10 | 85.40 |
| | IMP+SDT($\lambda = 0.0$) | VGG | 10 | 83.70 |
| | IMP+TET($\lambda = 0.0$) | VGG | 10 | 85.90 |
| | IMP+TET-S($\lambda = 0.0$) | VGG | 10 | **86.20** |
| | IMP+TET-S($\lambda = 0.2$) | VGG | 10 | **87.10** |
| | IMP+TET-S($\lambda = 0.4$) | VGG | 10 | **86.40** |
| ImageNet100 | TET | SEW-ResNet18 | 4 | 78.50 |
| | SDT | SEW-ResNet18 | 4 | 79.10 |
| | LTS | SEW-ResNet18 | 4 | **80.20** |
| | IMP+TET | SEW-ResNet18 | 4 | 78.70 |
| | IMP+SDT | SEW-ResNet18 | 4 | 79.90 |
| | IMP+LTS | SEW-ResNet18 | 4 | **80.80** |

Table 4 presents the results of a series of ablation studies conducted on the CIFAR10-DVS and ImageNet100 datasets, aimed at analyzing the impact of various factors on model performance. This helps understand the role of different components and parameters in the overall model, and aids in optimizing the model design. For the CIFAR10-DVS dataset, we explored methods including SDT, TET, and their variants TET-S and versions combined with IMP. VGG was used as the spiking neural network structure. The results show that for different $\epsilon$ values, the TET-S+IMP method achieved the best accuracy, with IMP+TET-S ($\lambda = 0.2$) reaching 87.10%, the highest among all methods on the CIFAR10-DVS dataset. For the ImageNet100 dataset, we tried TET, SDT, LTS, and their versions combined with IMP, using SEW-ResNet18 as the spiking neural network structure. On this dataset, the LTS method and its combined version with IMP, IMP+LTS, performed the best, reaching 80.80% accuracy.

## 6 Conclusions

We have proposed a learnable IMP by rethinking the membrane dynamics of SNNs to enhance the dynamics mechanism of spiking neurons. Additionally, we have presented a LTS post-processing method for the static tasks and a label-smoothed TET loss for the neuromorphic tasks. It is worth mentioning that our methods only require very minor modifications to the settings and loss functions of spiking neurons to effectively improve the performance of SNNs on the static tasks and the neuromorphic tasks. At the same time, almost no additional computational cost is required. Since our proposed method has broad compatibility with existing model structures and training methods, it can be widely applied on the existed methods to further improve their network performances.

## Acknowledgement

This work was supported by Fundamental Research Funds for the Central Universities (Grant No. SWU021002), Project of Science and Technology Research Program of Chongqing Education Commission (Grant No. KJZD-K202100203), Key R&D Program of Zhejiang (2022C01048), and National Natural Science Foundation of China (Grant Nos. U1804158, 62376247, U20A20220, and 62334014).

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

# A Appendix / supplemental material

## A.1 Broader Impacts

This paper focuses on the fundamental research of spiking neural networks, with the goal of revealing the impact of membrane dynamics on the network and optimizing its performance. Generally, there are no negative societal impacts in this work.

## A.2 Limitations

IMP has a small gradient during training, which makes it sensitive to initialization (Figure 4). In addition, learnable IMP may lead to an excessive number of parameters, as it assigns initial states to each neuron, although this has the same computational cost as setting the IMP to 0. The advantage of LTS may reduce when the time step is set too large, due to the supervision is only applied at the last time step. Therefore, we only recommend using LTS on static tasks with time steps less than 8 (Table 8), which should be able to handle most situations. Additionally, the performance of the combination of LTS and the latest spike transformer technology is not yet clear. Furthermore, we have not found a unified loss function that can achieve superior performance on both static tasks and neuromorphic tasks, which remains a challenge in the current research.

## A.3 Convergence Speed

We compared the convergence speed of TET and SDT at different time steps (T=1,2,4,6,8,12,16,24,32). For static tasks, TET's convergence speed was lower than SDT's, and the difference in convergence speed diminished as the time step increased.

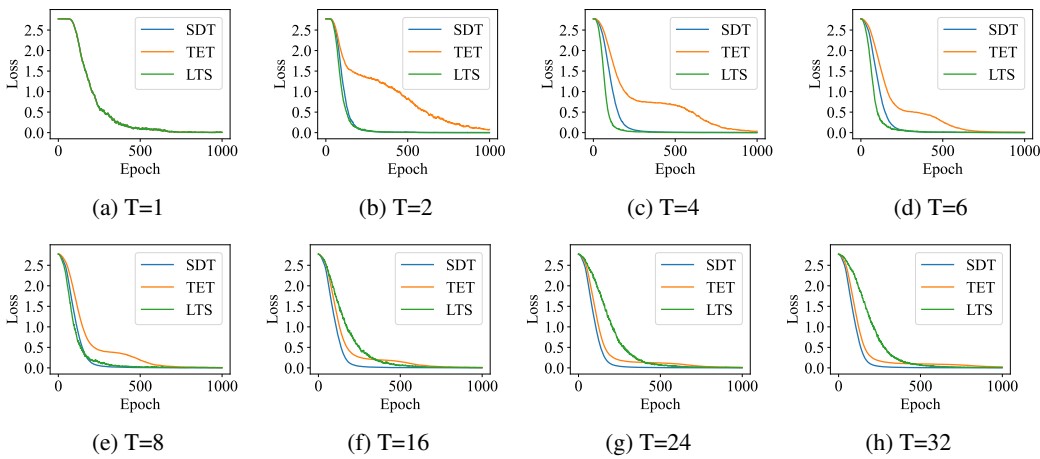

Figure 8: The convergence speed of SDT, TET and LTS on static data.

## A.4 Compared with Transformer-based SNNs

The performance of SEW-ResNet-LTS can be close to some Transformer-based SNNs (Table 5).

Table 5: Accuracy and theoretical energy consumption compared with Transformer-based SNNs.

| Model | Param (M) | SOPs (G) | Power (mJ) | Accuracy |
|---|---|---|---|---|
| Spikformer-8-384 | 16.81 | 6.82 | 7.73 | 70.24 |
| Spikformer-6-512 | 23.37 | 8.69 | 9.42 | 72.46 |
| Spike-driven 8-384 | 16.81 | - | 3.90 | 72.28 |
| Meta-SpikeFormer | 15.10 | - | 16.70 | 74.10 |
| SEW-R50-LTS (ours) | 25.56 | 3.10 | 2.79 | 71.24 |
| SEW-R50-LTS+IMP (ours) | 36.67 | 3.45 | 3.11 | 71.83 |

## A.5  Energy Consumption

IMP does not incur significant additional theoretical power consumption, but can effectively improve the performance of SNNs (Table 6).

Table 6: Accuracy and theoretical energy consumption on ImageNet1k dataset.

| Model | Training Method | Accuracy | SOPs(G) | Power(mJ) |
|---|---|---|---|---|
| SEW-ResNet18 | TET | 62.92 | 1.36055 | 1.22449 |
| SEW-ResNet18 | SDT | 63.21 | 1.37418 | 1.23676 |
| SEW-ResNet18 | LTS | 64.33 | **1.21427** | **1.09285** |
| SEW-ResNet18 | LTS+IMP | **65.38** | 1.31371 | 1.18234 |
| SEW-ResNet34 | TET | 67.98 | 3.59539 | 3.23585 |
| SEW-ResNet34 | SDT | 68.10 | 3.52732 | 3.17459 |
| SEW-ResNet34 | LTS | 68.10 | **3.11694** | **2.80525** |
| SEW-ResNet34 | LTS+IMP | **68.90** | 3.12180 | 2.80962 |
| SEW-ResNet50 | TET | 69.87 | 3.40181 | 3.06163 |
| SEW-ResNet50 | SDT | 70.33 | 3.20071 | 2.88064 |
| SEW-ResNet50 | LTS | 71.24 | **3.10432** | **2.79389** |
| SEW-ResNet50 | LTS+IMP | **71.83** | 3.45325 | 3.10792 |

## A.6  Performance of LTS on DVS Tasks

The LTS method can lead to information loss, especially on DVS tasks with a large number of time steps (Table 7). Therefore, we suggest considering the use of LTS only in static tasks, as the effectiveness of LTS relies on the assumption of having the same input at each time step.

Table 7: Accuracy on CIFAR10DVS dataset with different time-steps.

| Model | Training Method | T=4 | T=8 | T=10 | T=16 |
|---|---|---|---|---|---|
| VGG | TET | 83.8 | 85.0 | 85.8 | 86.4 |
| VGG | SDT | 83.4 | 84.3 | 84.4 | 85.1 |
| VGG | LTS | 83.7 | 83.0 | 82.9 | 82.3 |

## A.7  Enhancing Performance by Learnable IMP

The learnable IMP can significantly improve the accuracy of the first time step and lead to better overall performance (Figure 9).

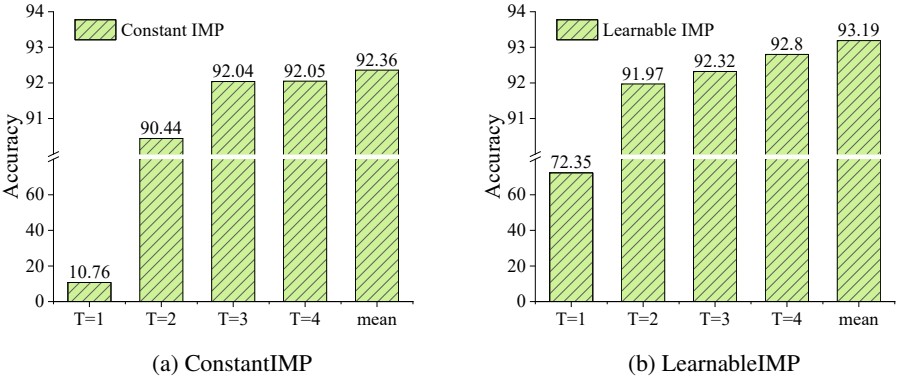

(a) ConstantIMP        (b) LearnableIMP

Figure 9: The test accuracy at each time step on the CIFAR10 dataset.

## A.8 Experimental Configurations and Hyperparameter Settings

Table 8 lists the key parameters required for training on the static datasets ImageNet1k, ImageNet100, CIFAR10, and CIFAR100. Table 9 outlines the key parameters used for training on the neuromorphic datasets CIFAR10-DVS-128, CIFAR10-DVS-48, N-Caltech101-128, and N-Caltech101-48.

Table 8: Experimental configurations on static task.

| hyper-parameter | ImageNet1K | ImageNet100 | CIFAR10 | CIFAR100 |
|---|---|---|---|---|
| architecture | SEW-ResNet | SEW-ResNet | SEW-ResNet | SEW-ResNet |
| time steps | 4 | 4 | 4 | 4 |
| enable TEBN | No | No | No | No |
| detach reset | Yes | Yes | Yes | Yes |
| spiking neuron | IF+IMP | IF+IMP | IF+IMP | IF+IMP |
| sg function | Atan | Atan | Atan | Atan |
| membrane decay | - | - | - | - |
| optimizer | AdamW | AdamW | AdamW | AdamW |
| learning rate | 0.001 | 0.001 | 0.001 | 0.001 |
| weight decay | 5e-4 | 5e-4 | 5e-4 | 5e-4 |
| momentum | - | - | - | - |
| epoch | 320 | 200 | 200 | 200 |
| warm up | 10 | 10 | 10 | 10 |
| lr schedule | cosine | cosine | cosine | cosine |
| loss function | SDT/TET | SDT/TET | SDT/TET | SDT/TET |
| label smooth | - | 0.1 | 0.1 | 0.1 |
| data augment | standard | standard | standard | standard |
| enable cutmix | No | Yes | Yes | Yes |
| enable mixup | No | Yes | Yes | Yes |
| GPUs | 4 | 1 | 1 | 1 |

Table 9: Experimental configurations on neuromorphic task.

| hyper-parameter | CIFAR10DVS-48 | CIFAR10DVS-128 | NC101-48 | NC101-128 |
|---|---|---|---|---|
| architecture | VGG11 | VGG11 | VGG11 | VGG11 |
| time steps | 10 | 16 | 10 | 16 |
| enable TEBN | Yes | No | Yes | No |
| detach reset | Yes | Yes | Yes | Yes |
| spiking neuron | LIF+IMP | LIF+IMP | LIF+IMP | LIF+IMP |
| sg function | ZIF | sigmoid | ZIF | sigmoid |
| membrane decay | 0.25 | 0.5 | 0.25 | 0.5 |
| optimizer | SGD | AdamW | SGD | AdamW |
| learning rate | 0.1 | 0.001 | 0.1 | 0.001 |
| weight decay | 5e-4 | 0.06 | 5e-4 | 0.06 |
| momentum | 0.9 | - | 0.9 | - |
| epoch | 200 | 200 | 150 | 150 |
| warm up | 0 | 30 | 0 | 30 |
| lr schedule | cosine | cosine | cosine | cosine |
| loss function | SDT/TET | SDT/TET | SDT/TET | SDT/TET |
| label smooth | 0.01 | 0.01 | 0.01 | 0.01 |
| event augment | standard | NDA | standard | NDA |
| enable cutmix | No | Yes | No | Yes |
| enable mixup | No | Yes | No | Yes |
| GPUs | 1 | 1 | 1 | 1 |

## A.9 On-chip Learning

The following approach can be useful for implementing on-chip learning IMP: (1) Use an auxiliary neuron to distribute IMP (by firing a spike) to the other neurons at the initial time step. (2) Optimize the synaptic weights of this auxiliary neuron to adjust IMP.

