# OpenReview forum: "Rethinking the Membrane Dynamics and Optimization Objectives of Spiking Neural Networks"
_NeurIPS.cc/2024/Conference — NeurIPS 2024 poster_

### Official Review · Reviewer_UfLQ · 2024-07-03

**Soundness:** 3
**Presentation:** 3
**Contribution:** 3
**Rating:** 6
**Confidence:** 4

**Summary:**

The paper discusses the role of initial membrane potential (IMP) of neurons in spiking neural networks (SNNs). The authors found that IMP has a significant impact on the firing patterns of LIF neurons. Then, they propose a learnable IMP mechanism to improve the performance of SNNs. Additionally, the paper introduces the last time step (LTS) approach to enhance convergence speed, and proposes a label smooth temporal efficient training (TET) loss to improve training. The empirical experiements on image classification tasks show the effeciveness of the propsoed methods.

**Strengths:**

1. The idea of learnable IMP is simple and intuitive, and easy to be implemented in computer simulations.
2. The paper is written in a very straightforward way, making it easy to follow.
3. The simulated experiments are comprehensive on image classifications tasks, showing consistent performance gain by using learnable IMP and additional techniques compared to vanilla SNNs.

**Weaknesses:**

1. One major concern is about whether learnable IMP can be implemented in neuromorphic chips since the IMP is a float number. To my experience, I do not believe it can be effectively implemented in current hardwares such as Loihi (and the whole paper does not discuss hardware implementation issues). If I am wrong, the authors please corrected me by comprehensively addressing this issue. If hardware implementation is hard or energy-ineffecient (so the main advantage of SNN disappears), then the proposed models should be compared with ANN models with similar model size (and apparently it is behind ANNs).

2. The title is info-less and not proper. The are a few problems. First, it should be the dynamics of neurons, not neuron networks, since the IMP is the property of individual neurons. Second, dynamics of neurons include various aspects, such as the model choice (LIF or Hudgekin-Hoxley model), noise term, etc. The readers obtain no information from "rethinking the dynamics", and other aspsects are not disucussed in the paper. A better title might be something like "learnable initial membrane potental enhances spiking neural networks".

3. By improving the dynamics of spiking neurons, it is reasonable to expect the model to be enhance in time series or sequential processing (e.g. NLP) tasks rather than image classifications tasks (since "dynamics" is temporal property). The lack of experiments on corresponding datasets making it unclear whether the proposed methods are effective only on CV tasks or in general.

4. When compared with baseline methods, the baseline methods should be shortly explained. Otherwise, it is unclear whether the comparisons are fair, and lack insights other than "our methods are good".

**Questions:**

- Can learnable IMP can be implemented in neuromorphic hardwares such as Intel Loihi?
- Can the authors estimate the theoretical energy consumption like in other SNN papers?
- How is the performance on other datasets such as time series or NLP?

**Limitations:**

The main limitation is about hardware implement, which the authors did not discuss.

---

> ### Author Rebuttal · Authors · 2024-08-06
>
> Thank you for your insightful feedback. We have carefully studied your comments and argue that your concerns can be addressed.
>
> > Weakness 1: One major concern is about whether learnable IMP can be implemented in neuromorphic chips since the IMP is  a float number. To my experience, I do not believe it can be effectively implemented in current hardwares such as  Loihi (and the whole paper does not discuss hardware implementation issues). If I am wrong, the authors please corrected  me by comprehensively addressing this issue. If hardware implementation is hard or energy-ineffecient (so the main  advantage of SNN disappears), then the proposed models should be compared with ANN models with similar model size (and  apparently it is behind ANNs).
>
> **Answer:** IMP can be implemented on neuromorphic chips. Loihi supports setting the membrane potential (a floating-point number) to a non-zero floating-point value, and the LAVA (Loihi's toolchain) also provides APIs for setting IMP.
>
> ```python
> import numpy as np
> from lava.proc.lif.process import LIF
>
> # Create LIF Neuron with constant Initial Membrane Potential (IMP)
> imp = 0.
> lif = LIF(
>     shape=(16,),   # Number and topological layout of units in the process
>     v=imp,         # Initial value of the neurons' voltage (membrane potential).
>     vth=1.0,       # Membrane threshold
>     dv=0.5,        # Inverse membrane time-constant
>     name="lif"
> )
>
> # Create LIF Neuron with custom Initial Membrane Potential (IMP)
> imp = np.random.uniform(high=0.2, low=-0.2, size=(16,))
> lif_imp_version1 = LIF(
>     shape=(16,),   # Number and topological layout of units in the process
>     v=imp,         # Initial value of the neurons' voltage (membrane potential).
>     vth=1.0,       # Membrane threshold
>     dv=0.5,        # Inverse membrane time-constant
>     name="lif"
> )
> ```
> It is worth noting that in previous research, IMP has always existed, but due to the lack of relevant research on more appropriate settings, researchers usually set it to a floating-point value of 0. Apart from that, according to our results on the ImageNet test set (**Table R1**), IMP does not incur significant additional  theoretical power consumption overhead, but can effectively improve the performance of SNNs.
>
>
> > Weakness 2: The title is info-less and not proper. The are a few problems. First, it should be the dynamics of  neurons, not neuron networks, since the IMP is the property of individual neurons. Second, dynamics of neurons include  various aspects, such as the model choice (LIF or Hudgekin-Hoxley model), noise term, etc. The readers obtain no  information from "rethinking the dynamics", and other aspsects are not disucussed in the paper. A better title might be  something like "learnable initial membrane potental enhances spiking neural networks".
>
> **Answer:**  We appreciate your suggestions. As you mentioned, in this paper, we have investigated the dynamics of spiking neurons and its impact on the supervised learning and output representations of SNNs. We have taken your suggestions into consideration and propose to change the title to "Enhancing Spiking Neural Networks via Initial Membrane Potential and Optimized Objectives".
>
> > Weakness 3: By improving the dynamics of spiking neurons, it is reasonable to expect the model to be enhance in time series or sequential processing (e.g. NLP) tasks rather than image classifications tasks (since "dynamics" is  temporal property). The lack of experiments on corresponding datasets making it unclear whether the proposed methods  are effective only on CV tasks or in general.
>
> **Answer:** Thank you for the suggestion to further validate the effectiveness of our method.  We have conducted additional experiments on time series tasks. By incorporating IMP in SNN-delay[1], we have achieved an effective improvement from 95.10% to 96.02% (**Table R2**). Nevertheless, it is important to note that the neuromorphic datasets we used in our experiments (page 8 **Table 2**), including CIFAR10DVS[1] and NCaltech101[2], contain temporal dynamics, rather than static images.
>
> [1] Li H, Liu H, Ji X, et al. Cifar10-dvs: an event-stream dataset for object classification[J]. Frontiers in neuroscience, 2017, 11: 309.
>
> [2] Orchard G, Jayawant A, Cohen G K, et al. Converting static image datasets to spiking neuromorphic datasets using saccades[J]. Frontiers in neuroscience, 2015, 9: 437.
>
> > Weakness 4: When compared with baseline methods, the baseline methods should be shortly explained. Otherwise, it is unclear whether the comparisons are fair, and lack insights other than "our methods are good".
>
> **Answer:**  Due to the length constraint of the main text, we have provided the ablation experiments in the appendix (page 15 **Table 4**). We kept the experimental setup and hardware equipment completely consistent in the ablation experiments, except the parameters that needed to be compared. We hope that the results of the ablation experiments can address your concerns, and we will pay more attention to the explanation of our experimental results, providing more insights.
>
> > Question 1: Can learnable IMP can be implemented in neuromorphic hardwares such as Intel Loihi?
>
> **Answer:** Yes. Please refer to the response to Weakness 1.
>
> > Question 2: Can the authors estimate the theoretical energy consumption like in other SNN papers?
>
> **Answer:** Please refer to the response to Weakness 1 and the PDF we submitted.
>
> > Question 3: How is the performance on other datasets such as time series or NLP?
>
> **Answer:** Please refer to the response to Weakness 3 and **Table R2**.
>
> > Limitation 1: The main limitation is about hardware implement, which the authors did not discuss.
>
> **Answer:** Please refer to the response to Weakness 1.

---

> > ### Comment · Reviewer_UfLQ · 2024-08-08
> > **Discussion**
> >
> > Thanks for the responses! I see how you can set the IMP of neurons to different values in Loihi by "imp = np.xxxxx". However, my question is whether IMP can be learnable via e.g., surrogate gradient back-prop in neuromorphic chips (as the paper proposed "learnable IMP).

---

> > ### Comment · Reviewer_UfLQ · 2024-08-12
> >
> > Dear authors,
> >
> > This is a kind reminder that I asked a followed-up question and the due of author-reviewer discussion is near. I am still a bit confused about the detailed implementations of learnable IMP (since I have never considered adjusting IMP on neuromorphic hardwares), and more importantly I am curious about this. If there are any detailed instructions in the paper or appendix that I have missed, please just point them out.
> >
> > Given that you have well addressed my other comments and the other reviewers' comments, I am wiling to raise my score if this question can be resolved.

---

> > > ### Author Response · Authors · 2024-08-12
> > >
> > > Thank you for your response. The primary focus of this paper is on enhancing the performance of SNNs by learnable IMP, and ensuring that the trained networks can be deployed on neuromorphic chips for low-power inference. Currently,  since neuromorphic chips [1] typically struggle to run standard surrogate gradient backpropagation [2,3], and synaptic plasticity-based approaches [4,5] often leads to lower performance，the performance of on-chip learning training SNNs still lags behind that of gradient-based methods [6,7]. Although some approaches have attempted to achieve on-chip approximate backpropagation [7-10], their training speed [11], network scale [12,13], and performance are still lower than standard surrogate backpropagation [2]. Therefore, in order to pursuit the SOTA performance and ensure energy efficiency, the related researches mainly focus on the approach of training high-performance SNNs [2,3,14,15] on GPUs and then deploying them on neuromorphic hardware [16].
> > >
> > > At present, although we are unable to provide a new concrete on-chip learning method for the learnable IMP within such a short time, we hope the following approach can be useful for realizing on-chip learning IMP：(1) Use an auxiliary neuron to distribute IMP (by firing a spike) to the other neurons at the initial time step. (2) Optimize the synaptic weights of this auxiliary neuron to adjust IMP. In addition, we will further discuss on-chip learning in the limitations section of this paper and explore it in our future work. Thanks again for your questions of this work, and we hope our response can address your concerns.
> > >
> > > [1] Davies M, Srinivasa N, Lin T H, et al. Loihi: A neuromorphic manycore processor with on-chip learning[J]. Ieee Micro, 2018, 38(1): 82-99.
> > >
> > > [2] Neftci E O, Mostafa H, Zenke F. Surrogate gradient learning in spiking neural networks: Bringing the power of gradient-based optimization to spiking neural networks[J]. IEEE Signal Processing Magazine, 2019, 36(6): 51-63.
> > >
> > > [3] Davies M, Wild A, Orchard G, et al. Advancing neuromorphic computing with loihi: A survey of results and outlook[J]. Proceedings of the IEEE, 2021, 109(5): 911-934.
> > >
> > > [4] Kheradpisheh S R, Ganjtabesh M, Thorpe S J, et al. STDP-based spiking deep convolutional neural networks for object recognition[J]. Neural Networks, 2018, 99: 56-67.
> > >
> > > [5] Zenke F, Ganguli S. Superspike: Supervised learning in multilayer spiking neural networks[J]. Neural computation, 2018, 30(6): 1514-1541.
> > >
> > > [6] Bal M, Sengupta A. Spikingbert: Distilling bert to train spiking language models using implicit differentiation[C]//Proceedings of the AAAI conference on artificial intelligence. 2024, 38(10): 10998-11006.
> > >
> > > [7] Yao M, Hu J, Hu T, et al. Spike-driven transformer v2: Meta spiking neural network architecture inspiring the design of next-generation neuromorphic chips[J]. arXiv preprint arXiv:2404.03663, 2024.
> > >
> > > [8] Kaiser J, Mostafa H, Neftci E. Synaptic plasticity dynamics for deep continuous local learning (DECOLLE)[J]. Frontiers in Neuroscience, 2020, 14: 424.
> > >
> > > [9] Lillicrap T P, Cownden D, Tweed D B, et al. Random synaptic feedback weights support error backpropagation for deep learning[J]. Nature communications, 2016, 7(1): 13276.
> > >
> > > [10] Bellec G, Scherr F, Subramoney A, et al. A solution to the learning dilemma for recurrent networks of spiking neurons[J]. Nature communications, 2020, 11(1): 3625.
> > >
> > > [11] Shrestha A, Fang H, Rider D P, et al. In-hardware learning of multilayer spiking neural networks on a neuromorphic processor[C]//2021 58th ACM/IEEE Design Automation Conference (DAC). IEEE, 2021: 367-372.
> > >
> > > [12] Renner A, Sheldon F, Zlotnik A, et al. The backpropagation algorithm implemented on spiking neuromorphic hardware[J]. arXiv preprint arXiv:2106.07030, 2021.
> > >
> > > [13] Shrestha A, Fang H, Wu Q, et al. Approximating back-propagation for a biologically plausible local learning rule in spiking neural networks[C]//Proceedings of the International Conference on Neuromorphic Systems. 2019: 1-8.
> > >
> > > [14] Fang W, Yu Z, Chen Y, et al. Deep residual learning in spiking neural networks[J]. Advances in Neural Information Processing Systems, 2021, 34: 21056-21069.
> > >
> > > [15] Zhou Z, Zhu Y, He C, et al. Spikformer: When spiking neural network meets transformer[J]. arXiv preprint arXiv:2209.15425, 2022.
> > >
> > > [16] Ziegler A, Vetter K, Gossard T, et al. Spiking Neural Networks for Fast-Moving Object Detection on Neuromorphic Hardware Devices Using an Event-Based Camera[J]. arXiv preprint arXiv:2403.10677, 2024.

---

> > > > ### Comment · Reviewer_UfLQ · 2024-08-12
> > > >
> > > > I appreciate the authors' response, which makes sense. I think hardware implementation is an important matter that worth deeper investigations. I raise my score to 6 accordingly.

---

### Official Review · Reviewer_Cj2j · 2024-07-10

**Soundness:** 3
**Presentation:** 3
**Contribution:** 3
**Rating:** 6
**Confidence:** 4

**Summary:**

This paper investigates how the initial membrane potential affects the neuronal spike pattern and the model performace. The evolve of initial membrane potential would generate novel firing pattern and furthermore change the SNN output. Thus, by making the initial membrane potential a trainable parameter, the SNN can achieve significant improvement.

**Strengths:**

1. This paper gives a detailed illustration that, by adjusting the initial membrane potential, additional pattern mapping can be generated, thus improving the expression capacity. So the proposed idea is natural.

2. This paper designs a specific label smooth loss function to effectively control the firing level, which is delicate and efficient.

3. Figure 4 shows that with a sufficient good membrane potential, one time step is enough for a good output, all these observations make the trainable initial membrane potential theoretical sound.

4. Enough experiments are done, which makes the proposed method more convinced. We can see with a trainable initial membrane potential, the accuracy are more satisfactory.

**Weaknesses:**

1. In this paper, all experiments and illustrations are 4 time steps SNN. It remains non clear that in the long time steps task, how much could the initial membrane potential influence the output.

**Questions:**

1. Since we only use the output of the last time step to give an inference, I wonder if there is any information missing in this process. Have you ever considered using the last few time steps output instead of just one?
2. From another point of view, adjusting the initial membrane potential is just injecting a controlled noise into the membrane potential at first. Have the authors ever considered giving a controlled noise to each time step to improve the performance?

**Limitations:**

Yes.

---

> ### Author Rebuttal · Authors · 2024-08-06
>
> Thank you for acknowledging our research direction and the innovative approach we have employed.
>
> > Weakness 1: In this paper, all experiments and illustrations are 4 time steps SNN. It remains non-clear that in the  long time steps task, how much could the initial membrane potential influence the output.
>
> **Answer:** To enhance the clarity and readability of the chapter content, we set T=4 to simplify the analysis process. It is worth noting that in the experimental section, we cover the cases of T values of 8, 10, and 16 (on page 8 Table 2). The experimental results demonstrate that IMP can maintain its effectiveness even at longer time steps (**Table R3** and **Table R11**). IMP also exhibits significant improvements in time series tasks, which can be seen in **Table R2**.
>
> |   Model    | Training Method | Spike Neuron | Time-step | Accuracy  |
> |:----------:|:---------------:|:------------:|:---------:|:---------:|
> | Spikformer |       TET       |     LIF      |    64     |   97.57   |
> | Spikformer |       TET       |  LIF + IMP   |    64     | **98.26** |
>
> **Table R10: Accuracy on DVS128Gesture dataset.**
>
> > Question 1: Since we only use the output of the last time step to give an inference, I wonder if there is any
> > information missing in this process. Have you ever considered using the last few time steps output instead of just one?
>
> **Answer:** In static tasks, using only the last time step can indeed lead to information loss, but applying the same supervision label to multiple or all time steps may also make the model difficult to fit. There may be a tradeoff between information loss and fitting difficulty. LTS only uses the last time step for supervision and output representation, which can directly verify the hypothesis.
> Additionally, we believe that using the mean value of a few time steps' outputs instead of just one could be a good alternative to LTS, especially when time steps is relatively large. As mentioned in the A.2 Limitations section, the advantages of LTS tend to weaken as T increases, particularly when $T > 8$, as shown in Figure 8.
>
> > Question 2: From another point of view, adjusting the initial membrane potential is just injecting a controlled noise
> > into the membrane potential at first. Have the authors ever considered giving a controlled noise to each time step to
> > improve the performance?
>
> **Answer:** We have not yet tried related methods, this is a direction that remains unexplored. Adding controlled noise at each time step can be seen as a new method of membrane potential reduction/increase, which may have the potential to improve the capacity of the model. Similar methods like GLIF have achieved beneficial improvements by learning key parameters in the membrane dynamics. In summary, we appreciate your suggestions for future research directions.

---

> > ### Comment · Reviewer_Cj2j · 2024-08-10
> > **Discussion**
> >
> > Thank you for the rebuttal. I'd like to stick to my original score.

---

### Official Review · Reviewer_jQZD · 2024-07-11

**Soundness:** 3
**Presentation:** 3
**Contribution:** 3
**Rating:** 6
**Confidence:** 5

**Summary:**

This paper analyzes the dynamics of membrane potential and proposes to improve performance by correcting the initial membrane potential to learnable parameters. This article proposes to use only the output of the last timestep as the classification feature during inference. In general, this paper proposes a simple but effective method.

**Strengths:**

1. This paper provides a comprehensive and detailed analysis of the dynamics of membrane potential, which can inspire others.
2. The method proposed in this paper is simple but effective. Taking SEW-ResNet as the basic model, IMP and LTS both bring significant gains.

**Weaknesses:**

Although this paper is well-analyzed, it still has some weaknesses.

It is not enough to just experiment with SEW-ResNet on a static dataset. If the residual technique is altered, such as MS-ResNet [1], the residual connection is calculated using the membrane potential. In this instance, will a better initialization of the membrane potential result in considerable gains? If the network design is altered, such as the Spikformer [2] with SEW residual connections and the Spike-driven transformer [3] with MS residual connections. Are the proposed IMP and LTS still valid?

[1] Hu Y, Deng L, Wu Y, et al. Advancing spiking neural networks toward deep residual learning[J]. IEEE Transactions on Neural Networks and Learning Systems, 2024.

[2] Zhou Z, Zhu Y, He C, et al. Spikformer: When Spiking Neural Network Meets Transformer[C]//The Eleventh International Conference on Learning Representations.

[3] Yao M, Hu J, Zhou Z, et al. Spike-driven transformer[J]. Advances in neural information processing systems, 2023, 36.

**Questions:**

1. In the DVS datasets, each timestep has its own unique information. Will LTS lose the information of the previous time step and cause unstable recognition accuracy?
2. Why does PSN lose dynamics and biological rationality? Aren't soft-reset SNNs dynamic? Biological rationality seems to have nothing to do with the reset method, and biological rationality is not important in deep learning.
3. Hard reset will cause the model to be unable to calculate in parallel. Is the method proposed in this article only applicable to Hard-reset SNNs? Can this method scale up to a larger timestep (for example, 64)? In language and autoregressive tasks, the timestep of SNN is aligned with the sequence length, which is extremely large. Using hard-reset SNNs may cause serious inefficiency. What do you think of this?

**Limitations:**

Please see the weaknesses and questions.

---

> ### Author Rebuttal · Authors · 2024-08-06
>
> Thank you for your insightful comments. We first list your advice and questions, then give our detailed answers.
>
> > Weakness 1: If the residual technique is altered, such as MS-ResNet, the residual connection is calculated using the membrane potential. In this instance, will a better initialization of the membrane potential result in considerable gains?
>
> **Answer:** Yes. The effectiveness of our proposed IMP and LTS methods in membrane potential residuals on the MS-ResNet architecture can be evaluated on the ImageNet100 dataset.
>
> |    Model    | Method  | Epoch |   Acc1    |   Acc5    |
> |:-----------:|:-------:|:-----:|:---------:|:---------:|
> | MS-ResNet18 |   TET   |  100  |   73.92   |   91.62   |
> | MS-ResNet18 |   SDT   |  100  |   74.78   |   91.98   |
> | MS-ResNet18 |   LTS   |  100  |   75.78   |   92.92   |
> | MS-ResNet18 | LTS+IMP |  100  | **76.12** | **93.36** |
>
> **Table R6: Accuracy on ImageNet100 dataset.**
>
> > Weakness 2: If the network design is altered, such as the Spikformer with SEW residual connections and the Spike-driven transformer with MS residual connections. Are the proposed IMP and LTS still valid?
>
> **Answer:** Yes, we have tried the SpikingResFormer[1] with self-attention mechanism and membrane potential residual connections. Due to the limitations of time and computational resources, we used fewer epochs compared to the original paper. Although LTS is slightly lower than SDT, it still outperforms TET, which is consistent with our analysis. Additionally, we have verified the effectiveness of the combination of self-attention mechanism and IMP on the CIFAR10DVS dataset (**Table R3** and **Table R8**). We hope these results can address your concerns.
>
> |        Model        | Method |  Accuracy  |   SOPs(G)   |  Power(mJ)  |
> |:-------------------:|:------:|:----------:|:-----------:|:-----------:|
> | SpikingResFormer-S  |  TET   |   73.500   |   3.77187   |   3.39468   |
> | SpikingResFormer-S  |  SDT   | **73.988** |   3.42255   |   3.08029   |
> | SpikingResFormer-S  |  LTS   |   73.974   | **3.31618** | **2.98456** |
>
> **Table R7: Accuracy and theoretical energy consumption on ImageNet1k dataset.**
>
> |   Model    | Training Method | Spike Neuron | Time-step | Accuracy |
> | :--------: | :-------------: | :----------: | :-------: | :------: |
> | Spikformer |       TET       |     LIF      |    16     |   82.8   |
> | Spikformer |       TET       |  LIF + IMP   |    16     | **83.4** |
>
> **Table R8: Accuracy on CIFAR10DVS dataset.**
>
> [1] Shi X, Hao Z, Yu Z. SpikingResformer: Bridging ResNet and Vision Transformer in Spiking Neural Networks[C]//Proceedings of the IEEE/CVF Conference on Computer Vision and Pattern Recognition. 2024: 5610-5619.
>
> > Question 1: In the DVS datasets, each timestep has its own unique information. Will LTS lose the information of the previous time step and cause unstable recognition accuracy?
>
> **Answer:** Yes, the LTS method can lead to information loss, especially in DVS tasks with a large number of time steps. We only suggest considering the use of LTS in static tasks, as the effectiveness of LTS relies on the assumption of having the same input at each time step.
>
> |   Model    | Training Method | T=4 | T=8 | T=10 | T=16 |
> |:----------:|:---------------:|:---:|:---:|:----:|:----:|
> |VGG| TET | 83.8 | 85.0 | 85.8 |86.4|
> |VGG| SDT | 83.4 | 84.3 | 84.4 |85.1|
> |VGG| LTS | 83.7 | 83.0 | 82.9 |82.3|
>
> **Table R9: Accuracy on CIFAR10DVS dataset (resize to 48).**
>
> > Question 2: Why does PSN lose dynamics and biological rationality? Aren't soft-reset SNNs dynamic? Biological  rationality seems to have nothing to do with the reset method, and biological rationality is not important in deep  learning.
>
> **Answer:** PSN is not a soft-reset dynamics, but rather a no-reset mechanism. PSN removes the reset mechanism and achieves parallelization through linear mapping or convolution (scanning) across the time dimension. But we agree with your point that biological rationality is not important in deep learning. We will modify the statement to: "It should be noted that the removal of the reset process in PSN means that the spiking activities in the previous time steps will not affect the membrane potential values in the subsequent time steps." and change the "Dynamic" item in the table to "Reset".
>
> > Question 3.1: Hard reset will cause the model to be unable to calculate in parallel. Is the method proposed in this  article only applicable to Hard-reset SNNs?
>
> **Answer:** Our proposed methods and theoretical analysis do not conflict with the reset mechanism. IMP can generate new spiking patterns and pattern mappings for both Hard and Soft reset SNNs. LTS is designed specifically for static tasks, and the reset mechanism used in SNNs does not affect our hypotheses and theoretical results.
>
> > Question 3.2: Can this method scale up to a larger timestep (for example, 64)?
>
> **Answer:** Yes,  IMP remains effective with larger numbers of time steps (T=64), as demonstrated by experiments on the CIFAR10DVS (**Table R3**)，and IMP is also effective on time series tasks, as shown in **Table R2**.
>
> > Question 3.3: In language and autoregressive tasks, the timestep of SNN is aligned with the sequence length, which is extremely large.  Using hard-reset SNNs may cause serious inefficiency. What do you think of this?
>
> **Answer:** We agree with your point that the hard-reset mechanism can lead to inefficiency and affect the performance of SNNs on time series tasks, as the information from previous time steps will be completely lost when hard-reset neurons fire.

---

> > ### Comment · Reviewer_jQZD · 2024-08-13
> >
> > Thank you for your response. Author's response addressed all my concerns. I will raise my score to weak accept.

---

### Official Review · Reviewer_AWRd · 2024-07-12

**Soundness:** 3
**Presentation:** 3
**Contribution:** 4
**Rating:** 7
**Confidence:** 4

**Summary:**

presents a novel approach to understanding and modeling the dynamics of spiking neural networks

**Strengths:**

offering new insights into SNN modeling and potential applications in various domains.
theoretical framework for SNN dynamics is novel and addresses existing limitations in the field.

**Weaknesses:**

Some of the assumptions in the theoretical framework could be more explicitly stated and justified.
Additional experiments, particularly in real-world scenarios, could further validate the proposed methods.
The introduction and related work sections could provide more context to better situate the contributions within the broader literature.

**Questions:**

Can the authors provide more details on the assumptions made in the theoretical framework? How do these assumptions impact the generalizability of the proposed methods?How does the proposed framework compare to other state-of-the-art models in terms of computational efficiency and scalability?

**Limitations:**

The authors have addressed the limitations of their work to a reasonable extent. They acknowledge the assumptions made in their theoretical framework and discuss potential areas for future research. However, the discussion on the broader societal impact of the work could be expanded, particularly in terms of ethical considerations and potential negative consequences.

---

> ### Author Rebuttal · Authors · 2024-08-06
>
> Thank you for your efforts in reviewing our article and providing constructive feedback. We’d like to reply to your concerns in detail.
>
> > Weakness 1: Some of the assumptions in the theoretical framework could be more explicitly stated and justified.
>
> **Answer:** We will provide a clearer explanation of our assumptions in the subsequent version and ensure that the proof process strictly follows the format specifications, in order to improve the accuracy and readability of the content.
>
> > Weakness 2: Additional experiments, particularly in real-world scenarios, could further validate the proposed methods.
>
> **Answer:** Thank you for the suggestion.  We can provide the performance of our IMP method on the DVS128Gesture dataset, which captures human gesture movement trajectories using event cameras, thus being closer to real-world scenarios. Due to the current limitations in time and computational resources, we hope to try applying our method on larger-scale real-world scenarios dataset in our future work.
>
> |   Model    | Training Method | Spike Neuron | Time-step | Accuracy  |
> |:----------:|:---------------:|:------------:|:---------:|:---------:|
> | Spikformer |       TET       |     LIF      |    64     |   97.57   |
> | Spikformer |       TET       |  LIF + IMP   |    64     | **98.26** |
>
> **Table R4: Accuracy on DVS128Gesture dataset.**
>
> > Weakness 3: The introduction and related work sections could provide more context to better situate the contributions within the broader literature.
>
> **Answer:** We agree with your suggestion. We will supplement the introduction and related work sections to ensure they can better highlight our contributions.
>
> > Question 1: Can the authors provide more details on the assumptions made in the theoretical framework?
>
> **Answer:** Sure. For static tasks, we define $f(\cdot)$ as the network computation, $\theta$ represents the network weights, $s[t]$ is the set of membrane potentials of all neurons in SNN with time step $t$, $x$ is the constant input intensity for time step $t=1,2,...,T$, and $y[t]$ is the corresponding output, the output of the SNN at each time step can be represented in the following form,
> $$
> y[t] = f(x, s[t], \theta).
> $$
> It can be found that the temporal variations of the output $y[t]$ is determined solely by the membrane potential set $s[t]$. Based on experimental observations, we have the following assumptions:
>
> **A1**: $s[t]$ is time-varying in SNNs.
> **A2**: The change in $s[t]$ can alter the output $y[t]$.
> **A3**: Applying the same supervised label to $y[t]$ across all time steps may lead to difficulties in SNN convergence.
>
> > Question 2: How do these assumptions impact the generalizability of the proposed methods?
>
> **Answer:** **A1** typically holds true, except in the absence of external input. For **A2**, these phenomena are commonly observed in experiments, although minor perturbations may not significantly alter the output of the SNN. For **A3**, it is important to note that for static image classification tasks, the correct results can be obtained without perfect fitting. Therefore, when the model has sufficient expressive power to output similar $y[t]$ for different $s[t]$, and the task is simple enough, the performance of TET can approach the level of LTS.
>
> > Question 3: How does the proposed framework compare to other state-of-the-art models in terms of computational efficiency and scalability?
>
> **Answer:** By combining our proposed method, the performance of SEW-ResNet can be close to the Transformer-based SNNs. However, compared to the current state-of-the-art Spiking Transformer models, there is still a certain gap (For more details, please refer to the PDF file we have submitted). Additionally, we hope to combine the proposed method with these advanced models, and explore its potential in a wider range of application scenarios.
>
> |      Architecture      | Param (M) | SOPs (G) | Power (mJ) | Accuracy |
> | :--------------------: | :-------: | :------: | :--------: | :------: |
> |    Spikformer-8-384    |   16.81   |   6.82   |    7.73    |  70.24   |
> |    Spikformer-6-512    |   23.37   |   8.69   |    9.42    |  72.46   |
> |   Spike-driven 8-384   |   16.81   |    -     |    3.90    |  72.28   |
> |    Meta-SpikeFormer    |   15.1    |    -     |   16.70    |  74.10   |
> |   SEW-R50-LTS (ours)   |   25.56   |   3.10   |    2.79    |  71.24   |
> | SEW-R50-LTS+IMP (ours) |   36.67   |   3.45   |    3.11    |  71.83   |
>
> **Table R5: Accuracy and theoretical energy consumption on ImageNet test set.**
>
> > Limitation 1: The discussion on the broader societal impact of the work could be expanded, particularly in terms of ethical considerations and potential negative consequences.
>
> **Answer:** In this work, we found that adjusting the initial membrane potential can alter the spiking patterns of SNNs. Furthermore, in static tasks, the variation in SNN membrane potentials can influence the output. This finding may lead to the development of novel adversarial attack methods targeting SNNs. Attackers could achieve this by manipulating the membrane potential at specific future time steps. Such attacks would only require perturbation of the first few input frames, and could control the timing of errors in the SNN at a future time step. Compared to adding adversarial noise at all input time steps, this approach could be more stealthy.

---

### Author Rebuttal · Authors · 2024-08-06

Thanks for all reviewers' valuable comments. We are encouraged that reviewers recognize the effectiveness of setting learnable initial membrane potential states (IMP) for spiking neurons, and consider the idea of using the output of the last time step for supervised learning and output representation in static tasks to be interesting. Meanwhile, most of the reviewers concerned about the power consumption of the learnable IMP method, as well as the effectiveness of the IMP method in time series tasks. Our responses to these questions are as follows.

### Power Consumption
IMP does not incur significant additional theoretical power consumption, but can effectively improve the performance of SNNs. More information can be found in the PDF we have submitted.

|    Model     | Method  | Accuracy  |   SOPs(G)   |  Power(mJ)  |
| :----------: | :-----: | :-------: | :---------: | :---------: |
| SEW-ResNet18 |   TET   |   62.92   |   1.36055   |   1.22449   |
| SEW-ResNet18 |   SDT   |   63.21   |   1.37418   |   1.23676   |
| SEW-ResNet18 |   LTS   |   64.33   | **1.21427** | **1.09285** |
| SEW-ResNet18 | LTS+IMP | **65.38** |   1.31371   |   1.18234   |
| SEW-ResNet34 |   TET   |   67.98   |   3.59539   |   3.23585   |
| SEW-ResNet34 |   SDT   |   68.10   |   3.52732   |   3.17459   |
| SEW-ResNet34 |   LTS   |   68.10   | **3.11694** | **2.80525** |
| SEW-ResNet34 | LTS+IMP | **68.90** |   3.12180   |   2.80962   |
| SEW-ResNet50 |   TET   |   69.87   |   3.40181   |   3.06163   |
| SEW-ResNet50 |   SDT   |   70.33   |   3.20071   |   2.88064   |
| SEW-ResNet50 |   LTS   |   71.24   | **3.10432** | **2.79389** |
| SEW-ResNet50 | LTS+IMP | **71.83** |   3.45325   |   3.10792   |

**Table R1: Accuracy and theoretical energy consumption on ImageNet1k dataset.**


### Time Series Tasks
We have conducted additional experiments on time series dataset Spiking Heidelberg Digits [1]. By incorporating IMP in SNN-delay [2], we have achieved an improvement from 95.10% to 96.02%, surpassing the current state-of-the-art ANN model Event-SSM [3], to our best knowledge.


|    Model     | Network Architecture | Spike Neuron | Param  | Accuracy |
| :----------: | :------------------: | :----------: | :----: | :------: |
| Event-SSM[3] |       SSM(ANN)       |      -       | ~400k  |  95.50   |
| SNN-Delay[2] |       MLP+DCLS       |     LIF      | 214.0k |  95.11   |
| SNN-Delay[2] |       MLP+DCLS       |  LIF + IMP   | 214.5k |  96.02   |

**Table R2: Accuracy on Spiking Heidelberg Digits.**

The Spiking Heidelberg Digits (SHD) [1] dataset is an audio-based classification dataset of 1k spoken digits ranging from 0 to 9 in the English and German languages. The audio waveforms have been converted into spike trains using an artificial model of the inner ear and parts of the ascending auditory pathway.

[1] Cramer B, Stradmann Y, Schemmel J, et al. The heidelberg spiking data sets for the systematic evaluation of spiking neural networks[J]. IEEE Transactions on Neural Networks and Learning Systems, 2020, 33(7): 2744-2757.

[2] Hammouamri I, Khalfaoui-Hassani I, Masquelier T. Learning Delays in Spiking Neural Networks using Dilated Convolutions with Learnable Spacings[C]//The Twelfth International Conference on Learning Representations.

[3] Schöne M, Sushma N M, Zhuge J, et al. Scalable Event-by-event Processing of Neuromorphic Sensory Signals With Deep State-Space Models[J]. arXiv preprint arXiv:2404.18508, 2024.

### Long Time-Steps

We further validated the effectiveness of IMP over longer time steps.

|   Model    | Training Method | Spike Neuron | Time-step | Accuracy |
| :--------: | :-------------: | :----------: | :-------: | :------: |
| Spikformer |       SDT       |     LIF      |    32     |   82.6   |
| Spikformer |       SDT       |  LIF + IMP   |    32     | **83.1** |
| Spikformer |       SDT       |     LIF      |    48     |   81.5   |
| Spikformer |       SDT       |  LIF + IMP   |    48     | **82.5** |
| Spikformer |       SDT       |     LIF      |    64     |   81.1   |
| Spikformer |       SDT       |  LIF + IMP   |    64     | **81.4** |

**Table R3: Accuracy on CIFAR10DVS dataset.**

---

### Decision · Program_Chairs · 2024-09-25

**Decision:**

Accept (poster)

**Comment:**

I have read all comments and responses. Reviews appear to be consentaneous, with four positive scores of 7, 6, 6, and 6. However, there are still some concerns that were not fixed, e.g., the hardware implementation presented by Reviewer UfLQ, which a significant point of spiking dynamics. Thus, I weakly recommend to accept this manuscript.